# Implications of tree expansion in shrubland ecosystems for two generalist avian predators

**A. C. Young**[1☯]*, **T. E. Katzner**[2☯], **D. J. Shinneman**[2☯], **T. N. Johnson**[3☯]

**1** Department of Fish & Wildlife Sciences, University of Idaho, Moscow, Idaho, United States of America, **2** U. S. Geological Survey, Forest and Rangeland Ecosystem Science Center, Boise, ID, United States of America, **3** Department of Fish & Wildlife Sciences, University of Idaho, Boise, Idaho, United States of America

☯ These authors contributed equally to this work.

\* aarony@nmsu.edu

**Data Availability Statement:** All relevant data are within the manuscript and its Supporting Information files. Data is also available at https://github.com/achristophery/Generalist-avian-predators-Young-et-al.-2023.

## Abstract

Shrublands globally have undergone structural changes due to plant invasions, including the expansion of native trees. Removal of native conifer trees, especially juniper (*Juniperus spp.*), is occurring across the Great Basin of the western U.S. to support declining sagebrush (*Artemisia* spp.) habitats and associated wildlife species, such as greater sage-grouse (*Centrocercus urophasianus*). One justification for conifer removal is that it may improve survival of sagebrush-associated wildlife by reducing the abundance of avian predators. However, the relationship between conifer expansion and predator distributions has not been explicitly evaluated. Further, although structural characteristics of habitat are important for generalist predators, overall prey abundance may also affect habitat use by predators. We examined habitat use of common ravens (*Corvus corax*) and red-tailed hawks (*Buteo jamaicensis*), two generalist predators whose populations are increasing in western North America, to variation in structural characteristics and prey distributions in sagebrush habitat that has experienced conifer expansion. Structural characteristics of habitat were important predictors of habitat use for both ravens and red-tailed hawks, whereas measures of prey abundance were unimportant for both species likely because generalist predators can use a wide variety of food resources. Ravens, but not red-tailed hawks, responded positively to increasing cover of juniper and the probability of habitat use was highest (> 0.95) where juniper cover within 100 m was > 20%. Habitat use by red-tailed hawks, but not ravens, was greater near cliffs but was not associated with juniper cover. Our study suggests that the removal of conifer in similar environments may lower the probability of habitat use for ravens, a common predator with significant impacts on many prey species. Therefore, we suggest conifer removal may improve sage-grouse reproductive success and survival depending on responses to conifer removal from other predators. Our results may be reflective of similar changes in rangeland ecosystems around the world undergoing expansion of conifer and other woody vegetation. Though species identities differ from sagebrush habitats, generalist avian predators in other habitats may have similar relationships with structural resources.

**Funding:** This research was supported by the Great Basin Landscape Conservation Cooperative and U. S. Fish and Wildlife Service, award # F16AC01182 (TJ), National Institute of Food and Agriculture, U. S. Department of Agriculture, McIntire Stennis project 1009779 (TJ), the Palouse Audubon Society (AY), and the University of Idaho College of Natural Resources Travel Award (AY). The funders had no role in study design, data collection and analysis, decision to publish, or preparation of the manuscript. Any use of trade, firm, or product names is for descriptive purposes only and does not imply endorsement by the U.S. Government. https://lccnetwork.org/lcc/great-basin https://www.nifa.usda.gov/grants/programs/capacity-grants/mcintire-stennis-capacity-grant.

**Competing interests:** The authors have declared that no competing interests exist.

## Introduction

Changing habitat structure from expansion of native plants or invasion by non-native plants is a global phenomenon that has implications for fauna. Many arid, semi-arid, and Mediterranean-type ecosystems are affected by the recent expansion of woody plants attributed to land use (especially livestock grazing), fire suppression, and climate change [1, 2]. Habitat changes have significantly affected distributions of wildlife species around the world, and generalist predators in particular have benefited from novel resources in altered habitats [3, 4]. As predator/prey interactions are altered, direct and indirect effects of habitat on prey demography can interact to compound the effects of habitat change on wildlife populations [5]. For example, expansion or invasion of plants can directly reduce resources such as food or shelter for a species while indirectly increasing predation risk by subsidizing predator populations [6–9]. Given the impacts that altered vegetation structure can have on wildlife populations, understanding the effects of these alterations on the predator community is vital for conserving imperiled prey populations.

Habitat structure is an important feature that can influence habitat use for taxa such as avian predators [10–13]. For these species, structural resources such as trees, cliffs, or utility poles may provide nesting substrates or protection from predation [14, 15]. Structural resources may also facilitate hunting strategies, either as concealment for ambush hunters or as perches for visual hunters [16–18]. For example, tree planting in Israel has led to population declines for an endangered lizard, in part because avian predators spend more time hunting in areas with vertical perches [19, 20]. In the Mediterranean, expansion of conifer in shrublands has been shown to lead to reductions in songbird nest success due to the increased presence of a generalist corvid species [21]. For avian predators that may utilize a wide variety of prey items, structural resources may be a primary factor influencing habitat use [22].

Although prey abundance is an integral part of habitat selection theory, it has rarely been incorporated into studies of habitat use by avian predators (but see [11]). As a proxy for prey abundance, studies often evaluate the effect of physical landscape features under the assumption that these structures influence the distribution of prey as well as hunting efficiency for avian predators (e.g., [15, 23]). However, studies that have directly tested the effect of prey abundance on habitat use by avian predatorshave found either that an interaction between habitat structure and prey abundance affect habitat use [24–26], or that habitat use is influenced by habitat structure alone (e.g., [11, 27]). Understanding the relative effects of both structural and prey resources on habitat use by avian predators may improve our understanding of predator response to landscape change in ecosystems experiencing the broad-scale expansion of woody plants.

The sagebrush steppe ecosystem in the western United States has received much recent conservation focus due to concern over the status of the greater sage-grouse (*Centrocercus urophasianus*, hereafter "sage-grouse") and other species associated with sagebrush habitat. Sage-grouse populations are broadly distributed but have declined range-wide [28, 29]. Since Euro-American settlement of the western United States, portions of the sagebrush (*Artemisia* spp.) biome have experienced an expansion of conifer trees, mostly pinyon pine (*Pinus* spp.) and juniper (*Juniperus* spp.), particularly in the Great Basin [30, 31]. Conifer expansion is a regional threat to sage-grouse because population declines have been attributed in part to the expansion of conifers into sagebrush habitat that was previously treeless [32]. Anecdotal evidence suggests that conifer expansion may affect predator-prey dynamics because survival rates for sage-grouse that use habitat featuring conifers are lower than those for sage-grouse in areas with no conifers [33]. However, few studies have directly tested the effect of conifer expansion on habitat use by avian predators, or predators in general (but see [34, 35]).

In recent years, increased abundance in sagebrush habitat of common ravens (*Corvus corax*) and red-tailed hawk (*Buteo jamaicensis*), two highly generalist avian predators (hereafter "avian predators") of sage-grouse and other sagebrush-associated wildlife, have been attributed to human development [23, 36]. Ravens are a common predator of sage-grouse nests, and the removal of ravens has increased recruitment of sage-grouse [37, 38]. To our knowledge, no study has explicitly focused on the relationship between conifer expansion and associated habitat use by avian predators of sagebrush-associated species.

Given the negative effects of conifer expansion on sage-grouse populations, multiple conifer removal projects have been initiated in sagebrush habitats across the Great Basin [39]. One such project, the Bruneau-Owyhee Sage Habitat (BOSH) project, will remove juniper across ~250,000 ha [40]. Policies that have been recently enacted by the federal government will facilitate additional conifer removal in the Great Basin by minimizing environmental review requirements for projects [41]. However, critical information gaps exist concerning wildlife responses to conifer removal in sagebrush habitat, especially responses of predators. If structural resources are the primary influence of habitat use for avian predators, then conifer removal may reduce the presence of avian predators and associated predation risk for prey. Conversely, if prey distributions are the primary influence of habitat use for avian predators, then potential increases in prey abundance following conifer removal may increase habitat use by avian predators in some sagebrush habitats. For example, the overall density of small mammals is lower in conifer woodlands than sagebrush habitat [42, 43]. If avian predators are insensitive to changes in habitat structure, costly efforts to remove conifer may be ineffective. Information about the primary factors that influence habitat use for avian predators in landscapes that have experienced conifer expansion may therefore help guide habitat restoration efforts for sagebrush-dependent wildlife.

To inform future restoration efforts, we tested the effects of habitat structure and prey distributions on habitat use for avian predators within a conifer removal area. Our goal for this study was to provide a first step towards explicitly evaluating the relationship between conifer expansion and removal and avian predators. We had two main objectives: 1) evaluate the relationship between habitat structures and habitat use by avian predators in sagebrush-juniper habitat, and 2) test for an effect of abundance of prey resources on habitat use by avian predators. We tested three hypotheses: 1.) structural resources are the primary influence on habitat use for avian predators, 2.) prey distributions are the primary influence on habitat use for avian predators, and 3.) there are additive or interactive effects between structural resources and prey distribution on habitat use by avian predators. We predicted that areas with greater conifer cover would exhibit an increased probability of habitat use by avian predators, but that prey distributions would further influence predator habitat use. Consequently, we also expected conifer removal to decrease habitat use by avian predators in the year following restoration.

## Methods

### Study site

Removal of juniper occurred in southwest Idaho, USA in the northern Great Basin (Fig 1) and began in August 2019. The study area is composed primarily of big sagebrush (*Artemisia tridentata*) and low sagebrush (*Artemisia arbuscula*) interspersed with one conifer species, western juniper (*Juniperus occidentalis;* hereafter, juniper). In areas classified as < 10% conifer cover at our study site, juniper has an average height of 2.7 m ± SD 2.1 and an average stem density of 19 trees/ha ± SD 25. In areas classified as >20% conifer cover at our study site, junipers have an average height of 3.6 m ± SD 2.3 and an average stem density of 198 trees/

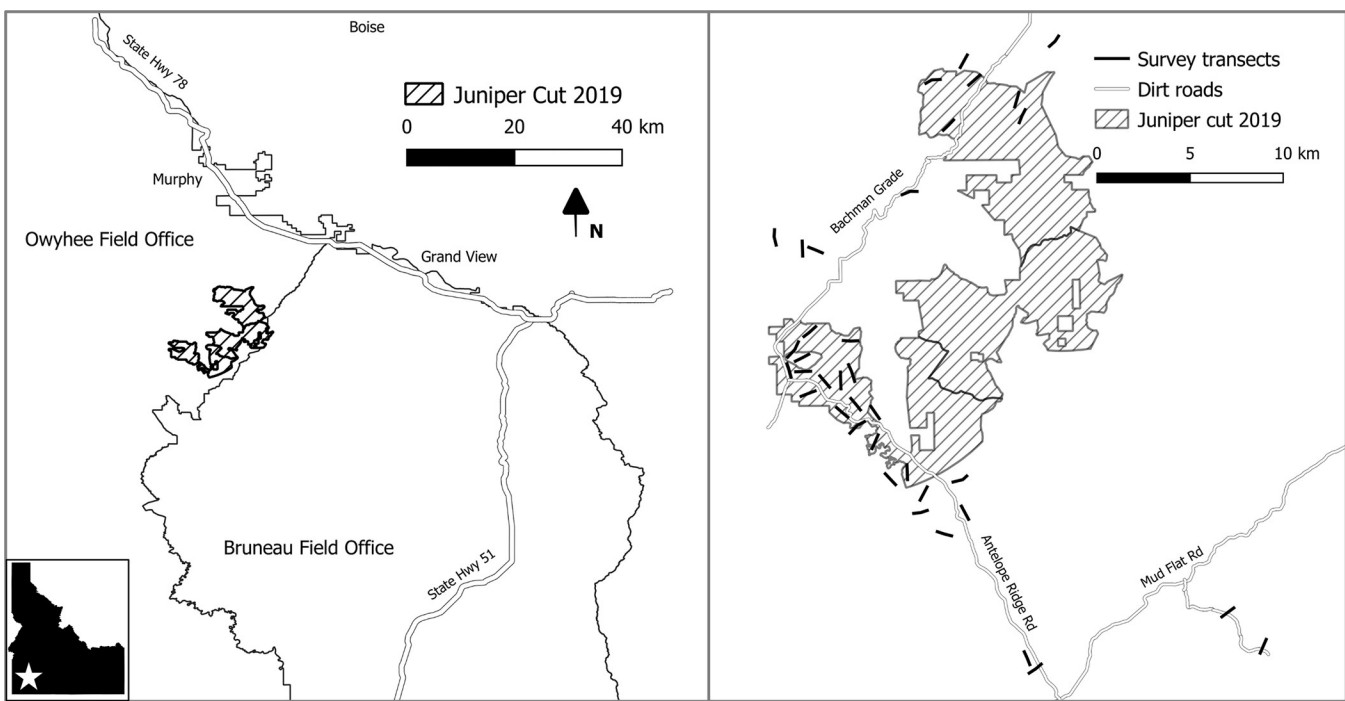

**Fig 1. Study site.** Location of avian predator surveys in the Owyhee Mountains of southwest Idaho, 2017–2020.

ha ± SD 193, but individual trees could reach heights of 12 m, and areal coverage of juniper can approach 60% [44]. Cheatgrass (*Bromus tectorum*) has invaded much of the study area, especially at lower elevations.

The climate of the study area is typified by hot summers and cold, snowy winters with an average of ~ 35 cm of precipitation annually [40]. Elevation ranges from 1,250 – 1,920 m and topography is varied with low-lying riparian areas interspersed with open sagebrush tablelands and rocky ridgelines. Before European settlement, juniper was most likely limited to rocky out-crops in small portions of the study area, presumably by historical fire regimes [45]. Since European settlement, juniper has expanded into sagebrush communities forming a gradient of tree cover across the landscape, and areas of dense juniper are currently found along ridgetops and in drainages [31]. Lesser, more scattered juniper cover typifies open sagebrush flats. Cattle grazing is ubiquitous across the study site, occurring at lower elevations during April-May and moving to higher elevations as summer progresses.

The BOSH project will eventually remove juniper cover classified as < 20% (at a 2-ha scale) from habitat formerly dominated by sagebrush across a ~250,000-ha landscape to support sage-grouse and other sagebrush-obligate species. Our sampling encompassed ~30,000 ha, and in the fall of 2019 15,000 ha of juniper was hand-cut within our study site. Juniper was cut using chainsaws and scattered so that no debris was higher than one meter.

## Raven and raptor counts

We conducted repeated-visit surveys for three years before and one year after juniper removal began. We surveyed 800-m transects (*n* = 37) to assess the effects of habitat structure on the probability of habitat use by avian predators within our study site. We surveyed each transect three times per year with at least two weeks between visits. We conducted surveys between May 1st and July 15th each year. We selected locations for survey transects using random points

in a GIS stratified by category of juniper cover (1–10%, 10–20%, and >20%) and location with respect to treatment plans for juniper removal (Fig 1). Eighteen of 37 transects were within areas where juniper would be removed. Surveys consisted of walking transects with three stationary, 10-minute observation periods placed at the beginning, middle, and end of each transect [23, 46]. Observers recorded any avian predators seen or heard while walking between or while at stationary observation points. We recorded birds if they were perched, calling, or flying/circling within the transect sample area. Each survey lasted ~ 45 minutes in total. Along each survey transect, the amount of time spent at stationary survey locations was consistent, but walking surveys between stationary survey locations varied based on terrain. We limited our data to observations of birds within 500 m of the observer to allow for more precise estimates of the effect of habitat on the probability of use and because this allowed us to assume that birds were influenced by the habitat surrounding the survey transect. We used laser range finders to estimate distances for birds detected visually and by sound. A single observer conducted each survey and three observers conducted 98% of surveys over four years. We conducted surveys before 10 am local time and never during steady rain or when estimated wind speeds exceeded 10 kph.

## Prey abundance

To test the effect of prey abundance on habitat use by avian predators, we estimated abundance, density, or presence for important prey groups. We considered the following measures for groups that are known to be common prey of both ravens and red-tailed hawks: densities of the most common species of small mammals including deer mouse (*Peromyscus maniculatus*) and Great Basin pocket mouse (*Perognathus parvus*); the relative abundance of songbirds; and ground squirrel presence/absence. Belding's ground squirrels (*Urocitellus beldingi*), which occur in large semi-colonial populations, are the most common ground squirrel at our study site.

To estimate the density of small mammals, we deployed 740 traps at five of the avian-predator transects in 2017 and 896 traps at the same five avian-predator transects in 2018 and 2019. We selected trapping locations for small mammals that represented a gradient of juniper cover, cheatgrass cover, and shrub structure (see [44]). In 2017, we used one trap array that consisted of 148 traps at the center of the avian predator transect. In 2018 and 2019, we used three trap arrays of 64 traps each at the center and ends of avian predator transects. We trapped small mammals over nine days broken up into four- and five-day sessions one month apart.

To estimate songbird abundance, we conducted 1,269 point-count surveys over four years along our avian-predator transects. Each survey transect for avian predators had three point counts stations placed at either end and in the middle of the survey transect. We surveyed songbirds concurrent with avian-predator surveys. We conducted 10-minute point counts three times per year at each avian-predator transect. We limited songbird observations to within 100 m of the point count station. We conducted songbird surveys from sunrise to 10 am and never during steady rain or winds stronger than 10 kph.

We noted the presence or absence of ground squirrels within 100 m of a survey transect because ground squirrels are an important food resource for red-tailed hawks. Ground squirrel presence is therefore treated as a categorical predictor variable.

## Prey measures

We used measures of individual prey groups as predictor variables to assess the importance of prey for habitat use by avian predators. We estimated the density of small mammals in

response to habitat characteristics at transect locations using a spatially explicit capture-recapture design (see [44]). We predicted small mammal density using the top-ranked models for both deer mice and pocket mice for sites where we did not sample mammals. We used vegetation measurements taken at each raptor-raven survey transect to predict small mammal density. The top model used to predict deer mouse density included a quadratic term for juniper cover within 100 m of the survey transect. The top model used to predict pocket mouse density included a quadratic term for mean sagebrush cover. We recorded the presence of ground squirrel colonies on survey transects using a categorical presence/absence approach. Ground squirrel presence on a survey transect was consistent across years, so each survey transect received a single classification for ground squirrel presence.

To calculate an index of songbird abundance, we pooled observations of all individuals and species and estimated the mean relative abundance of songbirds at the three points per transect. We did not adjust songbird observations for detection probability because differences in detectability among species would bias our abundance estimates, and previous studies examining the influence of songbird abundance on habitat use by avian predators have also used unadjusted relative mean counts [11].

## Habitat characteristics

We focused our sampling and analysis on natural and topographic features that may influence habitat use by avian predators. We also included anthropogenic landscape features in our data collection (e.g., cabins, roads) because the effects of human subsidies such as artificial vertical structures, roads, and food are well-established as characteristics that influence the distribution of raptors and ravens. However, our study site was relatively free of human infrastructure compared to similar studies of habitat use by avian predators. (e.g., [15, 23]).

We classified juniper cover across our study site with imagery collected at a 1-m scale from the National Agriculture Imagery Program (NAIP). To identify juniper in the image, we conducted a supervised image classification in ArcGis [47]. Next, we manually corrected any misclassifications for each of our transects using visual inspection and ground truthing. We then calculated the area of juniper cover within 100, 250, 500, 750, 1000, 1500, 2000, 2500, and 3000 m of our survey transects. These distances span a range of reported movements during the breeding season for ravens (mean movement 570 m from nest, 6.6 km$^2$ nest territory) and 40 km$^2$ core use area for non-breeding individuals [15, 23, 48] and breeding red-tailed hawks ($\bar{x}$ = 3.88 km$^2$ nest territory [14]). Three of our survey transects featured juniper mixed with mountain mahogany. We included mountain mahogany in our juniper-cover layer because individual tree species are not discernible in our classification, and we are interested in the effect of vertical structure on habitat use by avian predators. Thus, we assumed that avian predators respond to both tree species similarly, and that any association with trees in sagebrush-dominated habitats is a function of structure and not tree species composition. We calculated distance to cliff from each transect by defining cliffs as areas with greater than a 60˚ slope. We used a 10-m resolution digital elevation model (DEM; U.S. Geological Survey), and our definition of cliff areas as > 60˚ slope captures a 90˚ cliff that is 17 m high, a 100˚ cliff that is 25 m high, and a 110˚ cliff that is 54 m high. We also included a measure of distance to water using a GIS stream layer in combination with ponds created for livestock. We calculated distance to improved road and distance to nearest human dwelling for each transect.

## Occupancy models

To assess the effect of habitat characteristics and juniper removal on habitat use by avian predators, we used Bayesian multi-season occupancy models [49]. Multi-season occupancy models

[50, 51] are an extension of single-season occupancy models that allow for the estimation of changes in occurrence probability between seasons (in this case, years) through the estimation of parameters of extinction and colonization probability. Multi-season occupancy models assume that occupancy of a survey location is closed during each year but may change between years. If there is the possibility that the closure assumption is violated, as often happens for highly mobile species, the occupancy estimator may instead be considered the probability of habitat use [50]. We used the auto-logistic formulation of the multi-season occupancy model [49] to allow for inference on the effects of habitat covariates on overall occupancy probability as opposed to a decomposition of occupancy into colonization and extinction parameters. Use of the auto-logistic formulation is suitable for a limited sample size of unique sites and puts the inference focus on occupancy probability for each site for each year, as opposed to colonization and extinction probability [51].

To account for the imperfect detection of avian predators, we tested covariates that can affect the probability of detection using leave-one-out cross-validation [52]. We allowed detection to vary by year for all models and tested the effect of a terrain roughness index (TRI) and time of year on detection. We did not include an observer effect because a single observer (ACY) conducted 344 of 409 avian predator surveys (85%). Because we were interested in testing the effect of juniper cover on habitat use by avian predators, we did not include juniper cover as a covariate on detection in our occupancy models. We assumed that juniper cover would not lower our ability to detect large, loud species such as ravens. However, to validate this assumption we tested an exploratory model that included juniper cover within 100 m of the transect as a covariate on detection and with no covariates on habitat use. This exploratory model was not included in our set of candidate models. We standardized all predictor values and used normally distributed, non-informative priors for all covariate parameters (mean = 0, precision = 0.01).

## Variable and model selection

Uncertainty about the spatial scale at which habitat features may influence ecosystem processes is common for ecological studies. As a result, researchers are often interested in testing the effect of a habitat covariate at several spatial scales. However, the inclusion of all habitat covariates at all potentially relevant scales can lead to models that are difficult to interpret [53]. We reduced the number of spatial scales for variables included in our models following the screening procedure recommended by Stevens and Conway [53]. We fit univariate multi-season occupancy models for each scale (100 m– 3,000 m) of juniper cover that we quantified. We also tested other juniper metrics at these scales, including clustering and proportion of landcover comprised of three cover categories (<10% cover, 10–20% cover, and >20% cover). We estimated clustering using a K-means nearest neighbor analysis in ArcGIS. We estimated proportion of juniper cover categories by categorizing 30 m$^2$ pixels based on our juniper cover classification. We then compared the predictive power of each model in the set with leave-one-out cross-validation. Preliminary results from our model set testing the effect of different spatial scales of juniper measurement on habitat use suggested that ravens respond most strongly to the proportion of >20% juniper cover category habitat within 100 m of a transect. (β = 2.83, 90% CrI 0.41, 6.16; 99% posterior direction). However, because we were also interested in the relationship between all categories of juniper expansion and habitat use by ravens, we used continuous percent cover within 100 m of a survey transect, which was also a competitive model, as a covariate in our occupancy model set (β = 1.32, 90% CrI 0.13, 3.02; 99% posterior direction, Table A1 in S1 File).

Once we determined the spatial scale and measurement of juniper that best predicted habitat-use probability for ravens and red-tailed hawks, we constructed a model set that included

**Table 1. Models used to test effects of habitat features on detection and probability of habitat use for common ravens (*Corvus corax*) and red-tailed hawks (*Buteo jamaicensis*) in southwest Idaho, 2017–2020.**

| Detection | Habitat Structure Models | Prey Models |
|---|---|---|
| Null | Null | Structure model (sm) |
| Time of year | Juniper Cover | Small mammals |
| Terrain roughness index | Distance to cliff | Songbirds |
| | Distance to water | Ground squirrels |
| | Distance to stream | Small mammals + sm |
| | Distance to road | Songbirds + sm |
| | Distance to human dwelling | Ground squirrels + sm |
| | Distance to cliff + juniper | Small mammals*sm |
| | Distance to cliff + distance to stream | Songbirds*sm |
| | Distance to road + juniper | All prey groups additive |
| | Distance to water + juniper | |
| | Distance to human dwelling + juniper | |
| | Small mammal density | |
| | Juniper Removal | |

combinations of habitat features in addition to juniper that may influence habitat use for avian predators (Table 1). We then ran each model for at least 300,000 iterations using JAGS called from R [54]. We assessed model convergence using r̂ values generated by JAGS [55] and model convergence using visual inspections of traceplots [56]. We compared the predictive power of each model in the set using leave-one-out cross-validation [57, 58] and assessed model fit using the Freeman-Tukey test statistic to generate a Bayesian *p*-value [59]. Models with *p*-values close to 0.5 adequately fit the data, while extreme values close to 0 or 1 indicate a lack of model fit due to over- or under-dispersion. Finally, for each covariate in a model, we calculated the posterior direction, which is the percentage of the posterior distribution that has either a negative or positive value, depending on the sign of the covariate point estimate [60]. We calculated the average probability of habitat use and average detection probabilities for each year. In the final stage of modeling, we carried forward our top habitat structure model and tested for additive or interactive effects of prey abundance and habitat structure on the probability of habitat use for avian predators (Table 1).

## Permits and ethics statement

Field work for this study was carried out under Idaho Fish and Game research permit 161213 (T.N.J.). Sampling of small mammals was approved by the Institutional Animal Care and Use Committee of the University of Idaho (protocol #2017–14).

## Results

### Effect of structure on habitat use

We completed 409 avian predator surveys on the 29 transects sampled from 2017 to 2020 and eight additional transects sampled from 2018 to 2020. In 2017, we completed three surveys at 21 transects, two surveys at 10 transects, and one survey at two transects (*n* = 76 surveys). In 2018–2020, we completed three surveys at each of 37 transects (*n* = 111 surveys/year). Mean juniper cover within 100 m of all survey transects (*n* = 37) was 7.5% (8.1% SD) before juniper removal and 5.2% (8.4% SD) after removal, representing an average reduction in juniper cover of 30.6% per 10 ha. At transects where juniper was removed in fall 2019, average juniper cover

**Table 2. Sampling effort for surveys of common raven (*Corvus corax*) and red-tailed hawk (*Buteo jamaicensis*) in southwest Idaho, 2017–2020.**

|  | Pre-removal 2017 | Pre-removal 2018 | Pre-removal 2019 | Post-removal 2020 |
|---|---|---|---|---|
| Transects sampled | 30 | 37 | 37 | 37 |
| Surveys completed | 76 | 111 | 111 | 111 |
| **Ravens** |  |  |  |  |
| # Observations[1] | 35 | 236 | 205 | 161 |
| % of surveys [2] | 32% | 64% | 57% | 43% |
| % of transects[3] | 50% | 89% | 91% | 70% |
| **Red-tailed hawks** |  |  |  |  |
| # detections | 17 | 36 | 22 | 20 |
| % of surveys | 18% | 21% | 24% | 16% |
| % of transects | 40% | 45% | 40% | 37% |
| **Transect juniper cover category (100 m)** |  |  |  |  |
| 0% | 5 | 5 | 6 | 17 |
| 1–10% | 16 | 21 | 20 | 12 |
| 10–20% | 5 | 6 | 6 | 3 |
| > 20% | 4 | 5 | 5 | 5 |

[1] Total observations per year. [2] Percentage of surveys where a species was detected. [3] Unadjusted percentage of transects where a species was detected.

within 100 m of treated transects decreased from 4% (4% SD) to 0.009% (0.02% SD), representing an average reduction in juniper cover of 99% per 10 ha (Table 2).

We observed interannual variability in the number and spatial distribution of detections for both avian predator species (Table 2). Raven habitat use was positively influenced by percent cover of juniper and proximity to water (Fig 2A, Table A2 in S1 File). As juniper cover increased within 100 m of a transect, the probability of habitat use for ravens increased (β = 1.74, 90% Crl 0.32, 3.76; 99.5% posterior direction), and the highest probability of habitat use for ravens was areas with > 20% juniper cover. Credible intervals for the effect of distance to

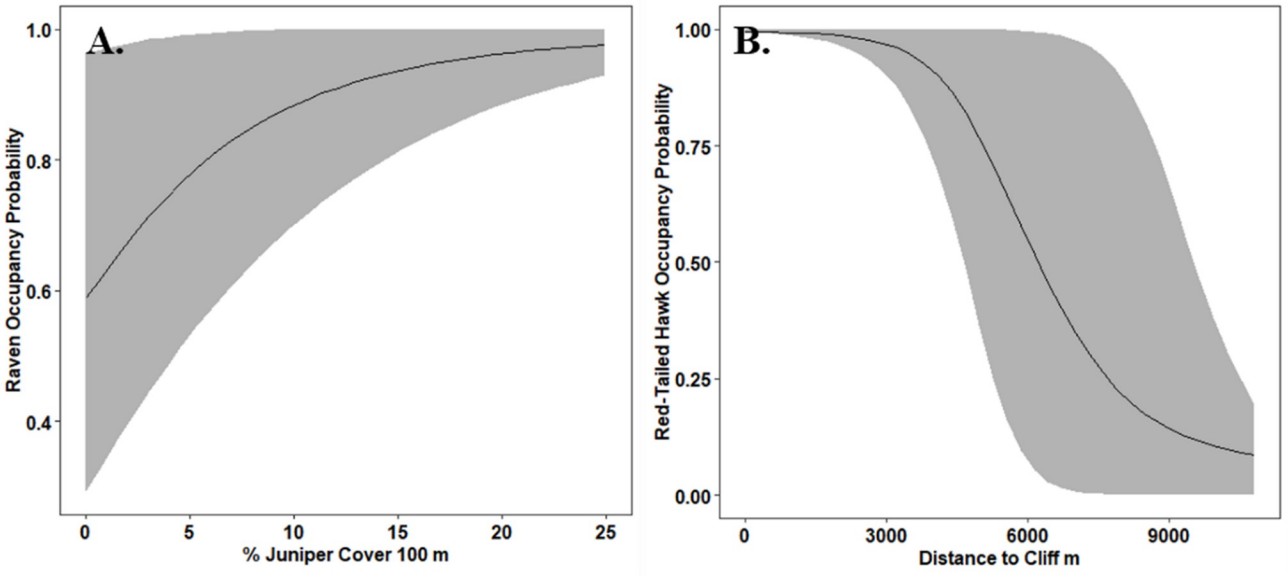

**Fig 2. Most predictive models.** Probability of habitat use for A.) common ravens (*Corvus corax*) and B.) red-tailed hawks (*Buteo jamaicensis*) in southwest Idaho, 2017–2020.

**Table 3. Derived parameters for the top-ranked common raven (*Corvus corax*) occupancy model in southwest Idaho, 2017–2020.**

| Parameter | Pre-removal 2017 | | Pre-removal 2018 | | Pre-removal 2019 | | Post-removal 2020 | |
|---|---|---|---|---|---|---|---|---|
| | Est. | SE | Est. | SE | Est. | SE | Est. | SE |
| [1]$\psi$ | 0.82 | 0.05 | 0.69 | 0.07 | 0.77 | 0.06 | 0.34 | 0.06 |
| [2]$p$ | 0.43 | 0.01 | 0.45 | 0.01 | 0.58 | 0.01 | 0.58 | 0.01 |
| [3] n.occ | 28.21 | 0.93 | 36.16 | 0.37 | 37.74 | 0.84 | 30.00 | 0.53 |

[1] average probability of habitat use, [2] mean detection probability, [3] estimated total number of used transects

water on habitat use by ravens overlapped zero (β = -0.65, 90% Crl -1.40, 0.03), but most of the posterior distribution for distance to water indicated that the probability habitat use by ravens declined as distance to water increased (95% posterior direction). The average probability of habitat use for ravens declined by 55% in the year following juniper removal (2019 = 0.77, SE 0.05, 2020 = 0.34, SE 0.06, Table 3). Posterior predictive checks and visual inspections did not indicate a lack of fit or lack of convergence for the most predictive model (Freeman-Tukey *p* value = 0.35, r̂ values = 1). We included TRI as a covariate for detection in our raven MSO models because TRI had the most predictive power for raven detection (β = 0.22, 90% Crl 0.008, 0.44; 90% posterior direction, Table A3 in S1 File). Juniper cover had a positive effect on the probability of detection of ravens (β = 0.43, 90% Crl 0.23, 0.59, 99% posterior direction, 0.12 Bayesian *p*-value). The probability of detection of ravens ranged from 0.43–0.58 (90% Crl range 0.27, 0.69, Table 3) over four years.

Probability of habitat use by red-tailed hawks was most strongly influenced by distance to a cliff, but differences in estimates of expected log predictive density (ELPD) suggest that this top model was only marginally more predictive than the null model. Probability of habitat use for red-tailed hawks declined as distance to cliff increased (β = -4.09, 90% Crl -7.57, -1.08; 97.3% posterior direction; Fig 2B, Table A4 in S1 File). Bayesian *p*-values suggest overdispersion in the models. We did not detect a statistically significant change in habitat use estimates for red-tailed hawks following juniper removal (Table 4). Red-tailed hawks responded most strongly to the clustering index of trees at a scale of 500 m, responding positively to more dispersed juniper. However, confidence intervals overlapped zero (β = 1.72, 90% Crl—4.82, 8.02; 77% posterior direction, Table A5 in S1 File). We did not identify important predictors for detection probability for red-tailed hawks (Table A6 in S1 File). Percent juniper cover did not have a strong effect on detection probability for red-tailed hawks (β = -0.15, 90% Crl -0.39, 0.09), which ranged from 0.18 − 0.24 over four years (90% Crl 0.10, 0.32).

## Effect of prey on habitat use

Estimates of prey abundance alone did not influence habitat use for either ravens or red-tailed hawks. For ravens, a model that included the relative abundance of songbirds along with

**Table 4. Derived parameters for the top-ranked red-tailed hawk (*Buteo jamaicensis*) occupancy model in southwest Idaho, 2017–2020.**

| Parameter | Pre-removal 2017 | | Pre-removal 2018 | | Pre-removal 2019 | | Post-removal 2020 | |
|---|---|---|---|---|---|---|---|---|
| | Est. | SE | Est. | SE | Est. | SE | Est | SE |
| [1]$\psi$ | 0.50 | 0.07 | 0.85 | 0.05 | 0.94 | 0.03 | 0.99 | 0.00 |
| [2]$p$ | 0.18 | 0.01 | 0.24 | 0.01 | 0.23 | 0.01 | 0.16 | 0.00 |
| [3]n.occ | 31.70 | 0.45 | 31.98 | 0.24 | 32.02 | 0.29 | 37.73 | 0.12 |

[1] average occupancy probability, [2] mean detection probability, [3] estimated total number of occupied transects

juniper cover ranked higher than the top-ranked model that included habitat structure only. However, as songbird abundance increased the probability of habitat use by ravens decreased (β = -0.80, 90% Crl -1.73, 0.12; Table A7 in S1 File) suggesting that ravens are less likely to use habitat with a higher abundance of songbirds. For red-tailed hawks, models including songbirds and small mammals as predictors of habitat use were ranked higher than distance to cliff alone (Table A8 in S1 File). However, increased songbird abundance was associated with a lower probability of habitat use for red-tailed hawks (β = -4.88, 90% Crl -9.17, -0.54), and credible intervals for the effect of small mammal density on habitat use by hawks widely crossed zero.

## Discussion

Our results provide evidence that structures such as trees and cliffs have a stronger influence on habitat use by generalist avian predators than the abundance of some of their most common prey resources. Higher juniper cover and proximity to cliffs increased the probability of habitat use by ravens and red-tailed hawks, respectively, while we found no effects of individual prey resources on habitat use by either species. Further, we found no evidence of an interaction between habitat structure and prey resources on habitat use by avian predators. For generalist predators such as ravens and red-tailed hawks, which have a high degree of diet plasticity, our results shed light on how habitat structure can influence habitat use whereas the abundance of any individual prey type may not. For example, in areas near agricultural fields, grains can make up the majority of raven diets [61], while in more natural types of vegetation cover, small mammals and songbirds can constitute the majority of raven diets [62]. Near roads, carrion is often consumed by both red-tailed hawks and ravens [26, 63]. Red-tailed hawks are also adaptable, with diet composition varying by region and prey abundance [64], and passerine birds and medium and small mammals including deer mice can be an important part of red-tailed hawk diets. Therefore, habitat structures that increase the probability of habitat use by generalist predators may increase the risk of predation for a wide range of prey species.

### Effect of structure on habitat use

The establishment of trees and other woody vegetation in rangeland habitats is occurring globally as a consequence of altered fire regimes, climate change, enhanced atmospheric $CO_2$, and livestock grazing [65, 66]. Conifer trees have expanded into shrub-steppe habitats in southern Canada [67], acacia (*Acacia saligna*) and pines (*Pinus* spp.*)* have become established in the fynbos shrublands of South Africa [68, 69], and pines have invaded the high Andean paramos ecosystem [70]. The establishment of trees in these systems has had negative impacts on some native wildlife associated with vegetation that existed before expansion. As the expansion of woody plants continues in other rangeland ecosystems across the globe, understanding how expansion affects distributions of both predator and prey species and subsequent predator-prey dynamics will be an important part of conserving grassland and shrubland-dependent species. Where habitat structure can be modified through the removal of expanding woody species, it will be important to assess whether there are limitations to the effectiveness of removal as a restoration technique, or if additional efforts are required. For instance, tree removal can increase cover of understory vegetation such as bunchgrasses and sagebrush shrubs [71, 72]. However, variation in soil condition, hydrologic factors, and the pre-removal dominance of conifer can limit the response of understory plants to conifer removal [73]. Further, variation in plant responses to tree removal techniques (i.e., mastication, hand cutting, etc.) may lead to the establishment of invasive annual grasses and contribute to differences in the recovery of faunal communities [74].

Effects of generalist predators are a widespread conservation concern for sensitive prey populations, and efforts to control predator populations can be difficult and controversial [75]. The conversion of sagebrush habitat to conifer woodlands has reduced and fragmented habitat for sagebrush-obligate wildlife, and increased abundances of generalist predators can compound the negative effects of habitat loss on prey species [76]. For example, conifer expansion has been linked to population declines for greater sage-grouse, and increased predator populations are suspected to play a role in these trends [32, 33]. However, evidence for an association between juniper expansion and numerical or functional responses of avian predators has been limited or speculative (e.g., [15, 23, 77]). Our results show that ravens are more likely to use sagebrush habitat experiencing conifer expansion, a relationship that has implications for the conservation of sagebrush-associated wildlife.

Subsidized populations of generalist predators can significantly impact prey populations because predation may continue even after the prey density becomes very low [78]. There are likely fewer anthropogenic subsidies at our study site than in many areas of the western U.S. given the low human density of Owyhee County, but raven populations more broadly have benefited from human development, leading to increased abundance in the Great Basin [79]. Housing density is low at our site (0.02 houses/km$^2$), and two lightly used dirt roads run through the study site (road density is 0.08 km/km$^2$). There are no agricultural fields within 10 km (distance from a transect to agriculture ranged from 10–43 km) and no transmission lines within 21 km (distance from a transect to a transmission line ranged from 21–55 km) of the study site. As a result, the effects of natural structures on the probability of habitat use are less likely to be confounded by the positive effects of human subsidies and structures. However, further research is needed to elucidate the relative effects of conifer woodlands and anthropogenic resources on the probability of habitat use for ravens. The addition of human structures to a landscape that also contains conifer woodlands may have an additive effect on the probability of habitat use for ravens. Ravens are 'incidental' hunters, consuming a wide variety of prey including small mammals, songbirds, and lizards [80]. As the amount of habitat used by ravens increases, the likelihood that ravens will incidentally prey on sage-grouse nests may also increase [36, 81].

For highly mobile species such as avian predators, violations of the assumption that animals remain in a survey location throughout a season may require that researchers consider the effects of habitat on an animal's presence and availability to be observed (i.e., "availability"; [50, 82]). Availability is defined as the probability that an animal is present at a survey location and able to be observed. Ravens are large, conspicuous birds that are easily seen and heard over long distances. Previous studies have sometimes assumed that observer error in detecting ravens is close to zero and estimated availability of ravens in relation to habitat, often referred to as use (e.g., [13]). For example, a study of ravens in the eastern United States estimated a 0.99 probability of detecting ravens over three visits to cliff locations if those locations were in fact occupied [83]. When the objective is to estimate the effects of habitat on an observer's ability to detect an avian predator, researchers most often test for effects of habitat features that can obscure avian predators, including trees or rough terrain. For example, O'Neil et al. [23] found support for using a 'viewshed index' that factored tree cover and TRI into an observer's ability to detect ravens. We found that juniper cover and TRI positively influenced detection of ravens, indicating that ravens are spending more time in habitat featuring juniper, regardless of whether or not ravens were being obscured from the observer by juniper or landscape features. At 0.43–0.58, our estimate of detection probability for ravens is higher than other studies (e.g., 35%, [23]). Given that we truncated our detections to 500 m, a smaller area than other studies (e.g., [23, 84]), we assume that our ability to see and hear ravens was high and that our estimate of detection probability represents availability to a large degree. Therefore,

the fact that both the probability of habitat use and availability are positively influenced by juniper cover provides further support for a relationship between juniper cover and habitat use by ravens. For red-tailed hawks, a relatively low detection rate in our data is likely a result of low density and home range sizes larger than the area of our survey transects. Low detection rates can bias occupancy estimates if there is unmodeled, non-random movement in and out of sampling units often associated with the breeding season or migration [50]. However, we tested a model with time of year as a covariate for detection and this model was not supported. When the closure assumption is violated but individual movements in and out of sampling units is random, the occupancy estimator is likely unbiased. In this case, occupancy estimates can be interpreted as probability of use [50].

## Effect of prey on habitat use

Similar to others, our study suggests that habitat structure is relatively more important than any specific prey resource for generalist avian predators [11, 27]. In fact, ravens were less likely to use habitat with higher abundances of songbirds and densities of small mammals. Densities of small mammals at our site, largely driven by deer mice, are highest at 10% juniper cover and decline as juniper cover increases [41]. Songbird abundance and diversity also increased in areas featuring early juniper expansion because the habitat can support sagebrush, ecotone, and conifer-associated songbirds [44, 67]. Longer-term data on prey-population distributions may reveal more complex predator-prey relationships that influence habitat use by generalist avian predators. It may also be that for generalist predators, no one prey population strongly influences habitat use. Data collected on prey populations at finer scales that account for within-season population variability and estimation errors associated with predictions for small mammal density and songbird abundance may improve inference. However, given the strong relationship we observed between habitat use for ravens and juniper woodlands, a disconnect between habitat use and prey abundance may be explained by the strong influence of vertical structure on habitat use (also reported in [23]).

## Management implications

Conifer removal may benefit wildlife species associated with sagebrush habitat by reducing habitat use for ravens, a common generalist predator. The relationship documented in this study between increasing tree cover in a historically tree-limited habitat and changes in habitat use for a generalist predator species has corollaries in shrubland and grassland ecosystems globally. For instance, an increased abundance of generalist predators has been shown to increase predation risk for small prey, such as in Australia where increased abundances of ravens and crows in rangelands decreased the abundance of shrub and ground-nesting songbirds [85]. Moreover, direct and indirect effects of woody-plant expansion on understory vegetation and interactions between trophic levels, respectively, may combine to alter rangeland wildlife communities and contribute to population declines of specialist species. As an example, changes to the structure of the small-mammal community can affect seed dispersal, potentially affecting vegetation structure and composition [86].

In our study, habitat use by ravens was most strongly influenced by juniper cover >20%, which is considered conifer woodland. This relationship has implications for habitat restoration efforts that focus primarily on the removal of conifer cover that is < 20% (e.g., BOSH) while allowing conifer cover >20% stands to remain intact. The relationship between habitat use by ravens and juniper cover < 20% was highly variable and far less predictive than the effect of juniper woodlands on habitat use. Sage-grouse avoid using habitat where the abundance of avian predators is high [87], and our findings suggest that riparian habitat lined with

juniper and near cliffs is likely to have the highest probability of use by avian predators at our site. Given that riparian habitat is important for juvenile sage-grouse during a vulnerable life stage, managing conifer expansion in and around riparian habitat may be particularly beneficial for sage-grouse populations. However, because we did not directly examine the impact of habitat-use by avian predators on sage-grouse demographics, future research on this topic will be an important next step for managers aiming to conserve populations of prey species.

Although areas with juniper cover < 20% are likely to have retained some shrub structure required by sagebrush wildlife, the association between ravens and juniper cover >20% suggests restoration efforts that do not remove dense juniper stands may not significantly alter the avian predator community. However, this is not to suggest removal of juniper cover < 20% may not have net benefits for sagebrush wildlife, as the mean probability of habitat use for ravens was lowest in the year following removal of juniper cover and would likely represent a net reduction in avian predator density for sagebrush wildlife. Moreover, logistical considerations are also important, as a primary objective of tree removal is to prevent the continued expansion of juniper [40]. Restoration of rangeland habitat dominated by late-stage conifer development may not be as successful as restoration of earlier successional stages [73], and removal of trees from shrublands can be costly and require long-term management through regular retreatment [88]. Finally, juniper woodlands are important for many wildlife species as well, and often feature high songbird diversity [39]. Though more research is needed on the implications of leaving juniper woodlands in sagebrush habitat, the benefits of removing juniper cover < 20% for sagebrush wildlife are important considerations for the conservation of sagebrush wildlife.

Given observed yearly variation in habitat use and abundance of avian predators, long-term monitoring, especially after tree removal, is required to better assess the effect of juniper removal on avian predators. An important next step will be to assess the effect of conifer removal on predation rates for prey species. However, to our knowledge, no long-term studies of habitat use by avian predators in a sagebrush-conifer woodland habitat exist. Therefore, our study provides early and valuable information on the overall effect of juniper cover on occupancy by avian predators.

## Supporting information

**S1 File. Model rankings for habitat use and detection.** Model rankings for the the effects of habitat structure and prey covariates on habitat use and the effect of habitat characteristics on detection of common raven (*Corvus corax*) and red-tailed hawk (*Buteo jamaicensis*) in southwest Idaho, 2017–2020.
(DOCX)

**S1 Data. Data and code.**
(ZIP)

## Acknowledgments

Our thanks to S. Copeland and B. Schoberle for logistical support and to our many field technicians for their hard work. Thank you also to S. Copeland for comments on a previous version of this manuscript. Any use of trade, firm, or product names is for descriptive purposes only and does not imply endorsement by the U.S. Government.

## Author Contributions

**Conceptualization:** T. E. Katzner, D. J. Shinneman, T. N. Johnson.

**Data curation:** A. C. Young.

**Formal analysis:** A. C. Young.

**Funding acquisition:** T. N. Johnson.

**Investigation:** A. C. Young.

**Methodology:** A. C. Young, T. E. Katzner, D. J. Shinneman, T. N. Johnson.

**Project administration:** T. N. Johnson.

**Supervision:** A. C. Young, T. N. Johnson.

**Writing – original draft:** A. C. Young.

**Writing – review & editing:** T. E. Katzner, D. J. Shinneman, T. N. Johnson.

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
