## [Decision Letter · Decision Letter 0]

9 Jan 2023

PONE-D-22-29377Structural resources and generalist avian predators: implications for tree expansion in shrubland ecosystemsPLOS ONE

Dear Dr. Young,

Thank you for submitting your manuscript to PLOS ONE. After careful consideration, we feel that it has merit but does not fully meet PLOS ONE’s publication criteria as it currently stands. Therefore, we invite you to submit a revised version of the manuscript that addresses the points raised during the review process.

Both reviewers acknowledge that the manuscript addresses an interesting question, although they have reasonable concerns about some aspects of the manuscript. I kindly invite the authors to try to address the methodological queries raised by reviewer 1, and to try to clarify the focus of the introduction and title of the manuscript as suggested by reviewer 2. 

We look forward to receiving your revised manuscript.

Kind regards,

Juan Manuel Pérez-García, PhD

Academic Editor

PLOS ONE

6. We note that Figure 1 in your submission contain [map/satellite] images which may be copyrighted. All PLOS content is published under the Creative Commons Attribution License (CC BY 4.0), which means that the manuscript, images, and Supporting Information files will be freely available online, and any third party is permitted to access, download, copy, distribute, and use these materials in any way, even commercially, with proper attribution. For these reasons, we cannot publish previously copyrighted maps or satellite images created using proprietary data, such as Google software (Google Maps, Street View, and Earth). For more information, see our copyright guidelines: http://journals.plos.org/plosone/s/licenses-and-copyright.

Reviewers' comments:

Reviewer's Responses to Questions

**Comments to the Author**

1. Is the manuscript technically sound, and do the data support the conclusions?

Reviewer #1: No

Reviewer #2: Partly

2. Has the statistical analysis been performed appropriately and rigorously? 

Reviewer #1: No

Reviewer #2: Yes

3. Have the authors made all data underlying the findings in their manuscript fully available?

Reviewer #1: Yes

Reviewer #2: No

4. Is the manuscript presented in an intelligible fashion and written in standard English?

Reviewer #1: Yes

Reviewer #2: Yes

5. Review Comments to the Author

Reviewer #1: This manuscript addresses an interesting management question and is based on a relatively large dataset. However, before it can be considered ready for publication I think it requires some work, both in its format and in some of the approaches/analyses.

I suggest that you phrase the hypotheses in a non-conditional way. E.g. “structural resources ARE the primary influence…”

The methods section requires much more detail and clarity. I would say that very few paragraphs would pass the test of successfully having a reader to replicate what you did. For instance, how many visits per year and per transect were done? Please explain. Also, during what season (or date ranges) was the field work done? What was the criterion for the inclusion of flying birds?, etc.

I do not understand the sentence “We did not adjust songbird observations for detection probability because differences in detectability among species would bias our abundance estimates…”. Precisely, adjusting for detectability tries to correct for a bias that is likely affecting your data. And in your case it could not only be related to species, sex and age of individual birds, but on habitat structure as well. Besides, simply because someone published a study without including such a correction, does not make it right. Because I assume that you did not record the distance at which individuals were detected, you may not be able to conduct this correction. So I suggest that you simply acknowledge that you did not do it, and later discuss the potential implications of this decision.

Please explain in detail how did you predict the abundance of small mammals for those transects where no trapping was conducted. I see a potential problem in this approach (i.e. using vegetation structure as a predictor of rodent abundance) since, later, you analyze both vegetation and prey abundance as, supposedly independent, predictors of predator occupancy.

The explanation for the procedure to generate the prey abundance score is very confusing, and, from what I could understand, rather arbitrary. The weights used for each attribute and value need to be better justified. Why, for instance, having presence of ground squirrels in the transect is equal to it being located at 5 km from the nearest lek? A more parsimonious way of integrating these different prey data could be using biomass per unit area. Probably you would have to make some assumptions for each case, but, at least you would have a more natural way of integrating these data.

If I understand it correctly, when you state “Multi-season occupancy models assume that occupancy of a survey location is closed during each year but may change between years.”, that means that the model considers that individuals are restricted to an area similar to that of the transect (i.e. 80 hectares) during a single season (year?). Although the home range of a hawk is likely much larger than that, I do not see a big problem if only one visit is made during the season. But if there are more than one visit during a year, then the closure assumption would be violated. As I mentioned in a previous paragraph, this information is not presented, but from my reading of the results section, it becomes evident that more that one visit was conducted in a year. I do not see a clear and easy way to address this problem, but it is certainly something that needs to be dealt with and, if possible, corrected.

I wonder to what extent the lack of patterns observed for red-tailed hawks is due to the spatial scale of the analysis. If you use a sampling area that is much smaller than the species’ territory or home range, then “occupancy” loses its meaning. First, because you likely have very few individuals to sample from, and, mostly because at some point they may leave the sampling area even for a few minutes or hours and you might miss them. You discuss this issue in lines 484-505, but I cannot understand why you conclude that in your case detection probability can be considered be equivalent to “availability”.

In order to rule out a significant effect of prey density on predator occupancy, it is important that this predictor is estimated with enough accuracy. In this particular study, different prey types (all with different methodological issues) were integrated in an index whose structure is not strongly justified. In addition, it is important to consider that in many cases, even though predators are likely to be attracted to areas with high abundance of prey, their very presence/abundance may reduce the carrying capacity for the prey, making more difficult the detection of such a relationship. You should be more cautious with your conclusions regarding this topic.

Reviewer #2: Review of Young et al.

This work assesses the relationship of habitat and prey availability with the occupancy of two generalist avian predators, common raven and red-tailed hawk. The authors analysed the habitat changes developed on an area that has suffered the elimination of conifer forest in order to compare it with areas that have not been eliminated, considering it to be an area of special interest for the conservation of the greater sage-grouse. The field work conducted, the variable selection and the methodology used are appropriate and provide great value to the manuscript. However, there are some aspects, mainly in the focus of the introduction, that could be improved for the understanding the results obtained in this study.

The authors introduce this manuscript with an focus that does not fit exactly with the context of the analyses performed and results obtained, since the introduction explains the effects of greater sage-grouse in relation to the removal of juniperus, due to the indirect effects that predators associated with juniperus forests may have. I consider that this relationship of causal effects is a more ambitious context than that which is subsequently analyzed in the models and the results obtained. I suggest focusing the introduction on what is strictly analyzed in methods, which is to address the factors that influence the occupancy of ravens and red-tailed hawks (e.g. L. 419-435). However, this can be contextualized that these results may have indirect implications for the greater sage-grouse, which has been explained in the "management implications" section.

L. 553-561. As indicated here, when more years of monitoring are available, the indirect relationship of predators on the greater sage-grouse due to juniperus management could be evaluated.

L. 105-117. Here, the authors focus this study in the sage-grouse, but the methodology is not developed in this way.

However, if the authors want to evaluate the grouse conservation problem, or the landscape management actions due to the elimination of conifers, they should specifically analyze the occupation of the grouse in relation to the % of junipers, as well as the abundance of predators, to know specifically if it is a problem of sage-grouse habitat or of predators. The juniper habitat may be removed, but the predators may still be there because they may be ecotone zones and the predators may have extensive hunting areas.

Other comments:

Title: I suggest a more concise and causal title: Implications of tree expansion in shrubland ecosystems on two generalist avian predators.

L. 33-37. I suggest to remove or summarize these sentences since I believe that it is not the main focus of this study.

L. 47. The latin name of western juniper should be previously in the abstract.

L. 53: ….”Therefor, we suggest that …” since it is not evaluated.

L.91. Please, include references about nest-trees as a resource for raptors and other predators.

L. 122-123. Rewrite this sentence if you refer to the relationship between forest expansion with predator occupancy and their indirect effects on prey, since specific effects of conifer exapansion and raptor occupancy could be for instance Jiménez-Franco et al. 2018. Plos One. Nest sites as a key resource for population persistence: A case study modelling nest occupancy under forestry practices.

L. 165: … of the study area…

L. 201-205: You shoukld justify the selected prey groups since in the introduction you only mention the sage-grouse.

L. 219: It should be the same unit that L. 197?

L. 246-253. Explain if this index is from previous work or is designed specifically for this study. Include an example of value or equation for its replication.

L. 323. Where is the results of the rest of models? Include them in Appendix.

L. 334-340. Include a table with these results.

Table 1. It should be explained if these variables are included in the same models, o different models.

L. 519: “ … generalist predators, the ravens”.

6. PLOS authors have the option to publish the peer review history of their article (what does this mean?). If published, this will include your full peer review and any attached files.

Reviewer #1: No

Reviewer #2: No

---

## [Author Response · Author response to Decision Letter 0]

6 Mar 2023

Reviewer 1:

I suggest that you phrase the hypotheses in a non-conditional way. E.g. “structural resources ARE the primary influence…”

Thank you, we made this change starting on line 151-154

The methods section requires much more detail and clarity. I would say that very few paragraphs would pass the test of successfully having a reader to replicate what you did. For instance, how many visits per year and per transect were done? Please explain. Also, during what season (or date ranges) was the field work done? What was the criterion for the inclusion of flying birds?, etc.

L189-192 Thank you, we more explicitly stated that we visited each survey location three times per year as well as the dates during which surveys were completed.

L. 197-198 We added language to clarify our approach. “Birds were recorded if they were perched, calling, or flying/circling within the transect sample area”.

See Anderson 2007 for raptor survey techniques including flying individuals

I do not understand the sentence “We did not adjust songbird observations for detection probability because differences in detectability among species would bias our abundance estimates…”. Precisely, adjusting for detectability tries to correct for a bias that is likely affecting your data. And in your case it could not only be related to species, sex and age of individual birds, but on habitat structure as well. Besides, simply because someone published a study without including such a correction, does not make it right. Because I assume that you did not record the distance at which individuals were detected, you may not be able to conduct this correction. So I suggest that you simply acknowledge that you did not do it, and later discuss the potential implications of this decision.

We added the following on L531:

“Data collected on prey populations at finer scales that account for within-season population variability and estimation errors associated with predictions for small mammal density and songbird abundance may improve inference.

”Please explain in detail how did you predict the abundance of small mammals for those transects where no trapping was conducted. I see a potential problem in this approach (i.e. using vegetation structure as a predictor of rodent abundance) since, later, you analyze both vegetation and prey abundance as, supposedly independent, predictors of predator occupancy.

L 240 Thank you, we revised to more explicitly explain our prediction method.

“We predicted small mammal abundance using the top-ranked models for both deer mice and pocket mice for sites where we did not sample mammals. We used vegetation measurements taken at each raptor-raven survey transect to predict small mammal density.”

 Juniper cover and prey abundance were not linearly correlated (r = 0.01 – 0.23) so we included them together in models. 

The explanation for the procedure to generate the prey abundance score is very confusing, and, from what I could understand, rather arbitrary. The weights used for each attribute and value need to be better justified. Why, for instance, having presence of ground squirrels in the transect is equal to it being located at 5 km from the nearest lek? A more parsimonious way of integrating these different prey data could be using biomass per unit area. Probably you would have to make some assumptions for each case, but, at least you would have a more natural way of integrating these data.

L 347 Table 1 We agree that the prey index and models including distance to lek may require too many assumptions. Therefore, we removed models that included the prey index and distance to lek from the candidate model set. Instead of combined prey index model, we added a global prey model that included songbird abundance + small mammal density + ground squirrel occurrence as predictor variables.

If I understand it correctly, when you state “Multi-season occupancy models assume that occupancy of a survey location is closed during each year but may change between years.”, that means that the model considers that individuals are restricted to an area similar to that of the transect (i.e. 80 hectares) during a single season (year?). Although the home range of a hawk is likely much larger than that, I do not see a big problem if only one visit is made during the season. But if there are more than one visit during a year, then the closure assumption would be violated. As I mentioned in a previous paragraph, this information is not presented, but from my reading of the results section, it becomes evident that more that one visit was conducted in a year. I do not see a clear and easy way to address this problem, but it is certainly something that needs to be dealt with and, if possible, corrected.

See revisions L 512-520: The reviewer is correct that the closure assumption may be violated, especially for red-tailed hawks. However, violations of the closure assumption are common for highly mobile species leading to a reinterpretation of the detection parameter as a combination of errors resulting from both the observer’s ability to observe the animal (observer error) and the availability of the animal to be seen (Gould et al. 2019). When the closure assumption is violated, it is more appropriate to interpret occupancy model estimates as “habitat use” (Mackenzie et al. 2018). If individuals randomly move in and out of a sample unit, MacKenzie et al. assert that the occupancy estimator is likely unbiased (p. 147). If individual movements are not random (as can happen during the breeding season or migration), covariates used to predict an effect on detection can limit this bias (MacKenzie et al. 2018, p. 148). We tested for an effect of time of year on detection for both species but this model was not supported. Moreover, a single visit to a transect within a season would preclude detection estimates within the mark-recapture occupancy framework (MacKenzie et al. 2009). 

There is strong support for the utility of occupancy models even when violations of the closure assumption occur (MacKenzie et al. 2009, Latif et al. 2016) 

I wonder to what extent the lack of patterns observed for red-tailed hawks is due to the spatial scale of the analysis. If you use a sampling area that is much smaller than the species’ territory or home range, then “occupancy” loses its meaning. First, because you likely have very few individuals to sample from, and, mostly because at some point they may leave the sampling area even for a few minutes or hours and you might miss them. You discuss this issue in lines 484-505, but I cannot understand why you conclude that in your case detection probability can be considered be equivalent to “availability”. 

See response above. 

L 512-520 “For red-tailed hawks, a relatively low detection rate in our data is likely a result of low density and home range sizes larger than the area of our survey transects. Low detection rates can bias occupancy estimates if there is unmodeled, non-random movement in and out of sampling units often associated with the breeding season or migration [48]. However, we tested a model with time of year as a covariate for detection and this model was not supported. When the closure assumption is violated but individual movements in and out of sampling units is random, the occupancy estimator is likely unbiased. In this case, occupancy estimates can be interpreted as probability of use [48]”.

”In order to rule out a significant effect of prey density on predator occupancy, it is important that this predictor is estimated with enough accuracy. In this particular study, different prey types (all with different methodological issues) were integrated in an index whose structure is not strongly justified. In addition, it is important to consider that in many cases, even though predators are likely to be attracted to areas with high abundance of prey, their very presence/abundance may reduce the carrying capacity for the prey, making more difficult the detection of such a relationship. You should be more cautious with your conclusions regarding this topic.

Thank you, we removed the prey index from our model set and added the following on L531-534

“Data collected on prey populations at finer scales that account for within-season population variability and estimation errors associated with predictions for small mammal density and songbird abundance may improve inference. 

Reviewer 2:

The authors introduce this manuscript with an focus that does not fit exactly with the context of the analyses performed and results obtained, since the introduction explains the effects of greater sage-grouse in relation to the removal of juniperus, due to the indirect effects that predators associated with juniperus forests may have. I consider that this relationship of causal effects is a more ambitious context than that which is subsequently analyzed in the models and the results obtained. I suggest focusing the introduction on what is strictly analyzed in methods, which is to address the factors that influence the occupancy of ravens and red-tailed hawks (e.g. L. 419-435). However, this can be contextualized that these results may have indirect implications for the greater sage-grouse, which has been explained in the "management implications" section.

Thank you for this helpful suggestion, and we agree that we do not address the effect of juniper removal on sage-grouse. However, we maintain that sage-grouse management is a primary driver for the funding for this work, and thus of primary interest for the management agencies who will use this work for inference (BLM 2018). 

L92-96 To clarify and broaden the context for our work, we added citations highlighting the effects of afforestation on predator-prey relationships in other parts of the world (Hawelena at al. 2006, 2010, Ben-David et al. 2019).

L. 557-565. As indicated here, when more years of monitoring are available, the indirect relationship of predators on the greater sage-grouse due to juniperus management could be evaluated.

We agree that further research directly assessing the effect of conifer removal on predator-prey relationships is needed. 

We added L146-147 “Our goal for this study was to provide a “first step” towards explicitly evaluating the relationship between conifer expansion and removal and avian predators.”

L. 105-117. Here, the authors focus this study in the sage-grouse, but the methodology is not developed in this way.

However, if the authors want to evaluate the grouse conservation problem, or the landscape management actions due to the elimination of conifers, they should specifically analyze the occupation of the grouse in relation to the % of junipers, as well as the abundance of predators, to know specifically if it is a problem of sage-grouse habitat or of predators. The juniper habitat may be removed, but the predators may still be there because they may be ecotone zones and the predators may have extensive hunting areas. 

Thank you for this helpful suggestion. On L 578-581 we added the following:

 “An important next step will be to assess the effect of conifer removal on predation rates for prey species.”

Though we agree that directly linking predator occupancy to sage grouse demographics and habitat use is an important next step, this type of analysis is beyond the scope of our study question and these data. Our goal for this study was to first examine the prediction that conifer expansion influences habitat use by avian predators. See Coates et al. 2020 and BLM 2018 for examples of connecting conifer expansion, increased abundance of avian predators, and concern for sage-grouse population status. The goal of our study was to be among the first to test the assumption that habitat use by avian predators is influenced by conifer expansion.

Title: I suggest a more concise and causal title: Implications of tree expansion in shrubland ecosystems on two generalist avian predators. 

Thank you, we have made this change.

L. 33-37. I suggest to remove or summarize these sentences since I believe that it is not the main focus of this study.

Thank you for this suggestion. We have retained this text because it is a primary justification for conifer removal in our study system and helps readers understand the motivation for our study (BLM 2018) 

L. 47. The latin name of western juniper should be previously in the abstract.

Thank you for this suggestion. Western juniper is not the only species at our study site, so it seems incorrect to only mention that species. However, we agree that it is important to mention the latin name and so we have added the generic name of junipers to the second sentence of the abstract.

L. 53: ….”Therefor, we suggest that …” since it is not evaluated.

Thank you, we made the suggested revision at line 51.

L.91. Please, include references about nest-trees as a resource for raptors and other predators.

L 90.We added the following additional reference:

Bosakowski et al. 1996

L. 122-123. Rewrite this sentence if you refer to the relationship between forest expansion with predator occupancy and their indirect effects on prey, since specific effects of conifer exapansion and raptor occupancy could be for instance Jiménez-Franco et al. 2018. Plos One. Nest sites as a key resource for population persistence: A case study modelling nest occupancy under forestry practices.

L 125-127 We clarified this sentence:

To our knowledge, no study has explicitly focused on the relationship between conifer expansion and associated habitat use by avian predators of sagebrush-associated species. L. 165: … of the study area…

We are unclear what the reviewer is suggesting.

L. 201-205: You should justify the selected prey groups since in the introduction you only mention the sage-grouse.

We added the following language explaining that these groups are common prey species for both ravens and red-tailed hawks in lines L 212-216:

“We considered the following measures for groups that are known to be common prey of both ravens and red-tailed hawks.”

L. 219: It should be the same unit that L. 197?

L 204 Thank you, we made this change.

L. 246-253. Explain if this index is from previous work or is designed specifically for this study. Include an example of value or equation for its replication.

We removed this combined prey index model from our model set per earlier comments.

L. 323. Where is the results of the rest of models? Include them in Appendix.

Thank you, we have included the complete model rankings in Appendix A

L. 334-340. Include a table with these results.

As suggested we have included all model rankings and p-values in Appendix A.

Table 1. It should be explained if these variables are included in the same models, o different models.

Table 1 L. 347 We revised the language in the table caption to clarify that these are the models we tested. It now reads: 

“Models used to test effects of habitat features on detection and probability of habitat use for common ravens (Corvus corax) and red-tailed hawks (Buteo jamaicensis) in southwest Idaho, 2017-2020.

L. 519: “ … generalist predators, the ravens”.

Thank you, we made this change on L 539:

“Conifer removal may benefit wildlife species associated with sagebrush habitat by reducing the occupancy of ravens, a common generalist predator.”

Literature cited

Andersen D.E. Survey techniques. Raptor research and management techniques. 2007:89-100.

BLM, 2018. Bruneau-Owyhee Sage-grouse Habitat Project (BOSH). DOI-BLM-ID-B000-2014-0002-EIS

Bosakowski, T., R.D. Ramsey, and D.G. Smith. 1996. Habitat and spatial relationships of nesting Swainson's Hawks (Buteo swainsoni) and Red-tailed Hawks (B. jamaicensis) in northern Utah. The Great Basin Naturalist, 56:341-347.

Bui T.V., J.M. Marzluff, B. Bedrosian. 2010. Common raven activity in relation to land use in western Wyoming: implications for greater sage-grouse reproductive success. The Condor, 112:65-78.

Coates P.S., S.T. O'Neil, B.E. Brussee, M.A. Ricca, P.J. Jackson, J.B. Dinkins, K.B. Howe, A.M. Moser, L.J. Foster, D.J. Delehanty. 2010. Broad-scale impacts of an invasive native predator on a sensitive native prey species within the shifting avian community of the North American Great Basin. Biological Conservation, 243:108409.

Conover M.R., A.J. Roberts. 2017. Predators, predator removal, and sage‐grouse: A review. The Journal of wildlife management, 81:7-15.

Gould, M.J., W.R. Gould, J.W. Cain III, and G.W. Roemer. 2019. Validating the performance of occupancy models for estimating habitat use and predicting the distribution of highly-mobile species: A case study using the American black bear. Biological Conservation, 234:28-36.

Harju, S. M., P.S Coates, S.J. Dettenmeier, J.B. Dinkins, P.J. Jackson, M.J. Chenaille. 2021. Estimating trends of common raven populations in North America, 1966-2018. Human-Wildlife Interactions, 15:248-269. 

Hawlena D, D. Saltz, Z. Abramsky, A. Bouskila. 2010. Ecological trap for desert lizards caused by anthropogenic changes in habitat structure that favor predator activity. Conservation Biology, 24:803-9.

Latif, Q. S., M.M. Ellis, and C.L. Amundson. 2016. A broader definition of occupancy: comment on Hayes and Monfils. Journal of Wildlife Management, 80:192–194.

MacKenzie, D.I., J.D. Nichols, M.E. Seamans, and R.J. Gutiérrez. 2009. Modeling species occurrence dynamics with multiple states and imperfect detection. Ecology 90:833.

MacKenzie D.I., J.D. Nichols, J.A. Royle, K.H. Pollock, L. Bailey, J.E. Hines. Occupancy estimation and modeling: inferring patterns and dynamics of species occurrence. Elsevier; 2017 Nov 17.

O'Neil ST, P.S. Coates, B.E. Brussee, P.J. Jackson, K.B. Howe, A.M. Moser, L.J. Foster, D.J. Delehanty. 2018. Broad‐scale occurrence of a subsidized avian predator: Reducing impacts of ravens on sage‐grouse and other sensitive prey. Journal of Applied Ecology. 55:2641-52.

---

## [Decision Letter · Decision Letter 1]

28 Apr 2023

PONE-D-22-29377R1Implications of tree expansion in shrubland ecosystems on two generalist avian predators .PLOS ONE

Dear Dr. Young,

Thank you for submitting your manuscript to PLOS ONE. After careful consideration, we feel that it has merit but does not fully meet PLOS ONE’s publication criteria as it currently stands. Therefore, we invite you to submit a revised version of the manuscript that addresses the points raised during the review process.

We are sorry to be late with your MS decision, but we were unable to contact one of the previous reviewers, so  we had to invite a new reviewer. 

Despite this, both reviewers are very positive and have only shown minor concerns that are easily addressed by you. 

We look forward to receiving your revised manuscript.

Kind regards,

Juan Manuel Pérez-García, PhD

Academic Editor

PLOS ONE

Journal Requirements:

Reviewers' comments:

Reviewer's Responses to Questions

**Comments to the Author**

1. If the authors have adequately addressed your comments raised in a previous round of review and you feel that this manuscript is now acceptable for publication, you may indicate that here to bypass the “Comments to the Author” section, enter your conflict of interest statement in the “Confidential to Editor” section, and submit your "Accept" recommendation.

Reviewer #2: All comments have been addressed

Reviewer #3: (No Response)

2. Is the manuscript technically sound, and do the data support the conclusions?

Reviewer #2: Yes

Reviewer #3: Yes

3. Has the statistical analysis been performed appropriately and rigorously? 

Reviewer #2: Yes

Reviewer #3: Yes

4. Have the authors made all data underlying the findings in their manuscript fully available?

Reviewer #2: No

Reviewer #3: Yes

5. Is the manuscript presented in an intelligible fashion and written in standard English?

Reviewer #2: Yes

Reviewer #3: Yes

6. Review Comments to the Author

Reviewer #2: The manuscript is clear and easy to foollow, it is now appropiate for publication. I have included only a couple of minor comments related to editing and to the replicability of this work in terms of its statistical analyses.

Line 170. Replace project by study area: "The climate of the study area is ...".

I suggest including a JAGS code in an appendix for the replicability of the work in other contexts. I believe that this information will add value to the manuscript, as it is free software that is being included in an increasing way of scientific articles and open source journals.

Reviewer #3: This is a well-written paper that uses a robust dataset to evaluate the implications of tree expansion over two generalist predator presence in a sagebrush ecosystem. Clearly a lot of work went into data collection, which includes predator species and his prey across a sagebrush steppe. The study has strong potential to be useful for management and conservation, but in my opinion there are some issues that need to be resolved before this paper is published.

The authors discuss potential implications of their study for the management and conservation of Sage-grouse but the data and results support this indirectly. Although data on the abundance of the potential prey of two generalist predators were used, there are no data on the abundance or presence of the Sage-grouse. In the same way, the authors argue that the abundance of prey does not influence the habitat use by the two species of predators, but only a part of the potential prey was studied. I suggest caution with the conclusions drawn from the results. Is it possible to predators are responding to another prey species present in the area?

Detailed comments

Introduction

Overall, the introduction is very well written and does a great job of setting the study.

L 96-97 delete space before “For”

You mentioned “For generalist predators that may utilize a wide variety of prey items, structural resources may be an important factor influencing habitat use.” But this also happens with forest-dwelling species and specialists. See Martínez-Hesterkamp S, Rebollo S, Pérez-Camacho L, García-Salgado G, Fernández-Pereira JM (2018) Assessing the ability of novel ecosystems to support animal wildlife through analysis of diurnal raptor territoriality.PLoSONE13(10):e0205799

Also, could you include a reference supporting those lines.

In the title and abstract you mentioned the term generalist predators but in the introduction only referred to avian predators, except in L. 96-97. This may be confused for readers. I suggest being consistently with one term for predators. In the objectives L. 148-155 you use avian predators not generalist predators. I suggest the use of generalist predators in the manuscript.

Methods

Please include references in the section of Raven and raptors counts.

L 198 How much time did you spend in stationary counts? This was equal in all transects? Please describe.

L 215 Please mention the ground squirrel species

L225-226 The transects for songbirds was an extension of 1.200 m, you mentioned that transects for avian predators had 800 m with 3 points counts within the transect in L 194-195. This it is not clear.

L 233-235 “Belding’s ground squirrels (Urocitellus beldingi), which occur in large semi-colonial populations, are the most common ground squirrel at our study site.” This could be used in the first paragraph of prey abundance. L. 214-216.

Discussion

L 555-560 I suggest being more cautious with these lines, the relationship between the study and the presence of the sage-grouse is not clear.

7. PLOS authors have the option to publish the peer review history of their article (what does this mean?). If published, this will include your full peer review and any attached files.

Reviewer #2: No

Reviewer #3: No

---

## [Author Response · Author response to Decision Letter 1]

10 May 2023

Reviewer #2: The manuscript is clear and easy to foollow, it is now appropiate for publication. I have included only a couple of minor comments related to editing and to the replicability of this work in terms of its statistical analyses.

Line 170. Replace project by study area: "The climate of the study area is ...".

Thank you, we made this change on line 172

I suggest including a JAGS code in an appendix for the replicability of the work in other contexts. I believe that this information will add value to the manuscript, as it is free software that is being included in an increasing way of scientific articles and open source journals.

Thank you, we have included JAGS model code in the data submitted to PLOS one during the resubmission process and this code will be available freely.

Reviewer #3: This is a well-written paper that uses a robust dataset to evaluate the implications of tree expansion over two generalist predator presence in a sagebrush ecosystem. Clearly a lot of work went into data collection, which includes predator species and his prey across a sagebrush steppe. The study has strong potential to be useful for management and conservation, but in my opinion there are some issues that need to be resolved before this paper is published.

The authors discuss potential implications of their study for the management and conservation of Sage-grouse but the data and results support this indirectly. Although data on the abundance of the potential prey of two generalist predators were used, there are no data on the abundance or presence of the Sage-grouse. In the same way, the authors argue that the abundance of prey does not influence the habitat use by the two species of predators, but only a part of the potential prey was studied. I suggest caution with the conclusions drawn from the results. Is it possible to predators are responding to another prey species present in the area?

Thank you for helpful comments, we appreciate your time and effort. We agree that our examination of the effect of prey distributions on predator occupancy may be limited by our ability to include all potential prey and that the effects of avian predators on sage-grouse demographics at our study site is unknown. However, because the removal of juniper at our study site was largely prompted by concerns over sage-grouse population conservation, we want to present our findings in the context of potential changes to the predator community for this species and highlight directions for future research. We added language in the discussion to this effect.

L 562. However, because we did not directly examine the impact of habitat-use by avian predators on sage-grouse demographics, future research on this topic will be an important next step for managers aiming to conserve populations of prey species.

Detailed comments

Introduction

Overall, the introduction is very well written and does a great job of setting the study.

L 96-97 delete space before “For”

Thank you, we corrected this formatting.

You mentioned “For generalist predators that may utilize a wide variety of prey items, structural resources may be an important factor influencing habitat use.” But this also happens with forest-dwelling species and specialists. See Martínez-Hesterkamp S, Rebollo S, Pérez-Camacho L, García-Salgado G, Fernández-Pereira JM (2018) Assessing the ability of novel ecosystems to support animal wildlife through analysis of diurnal raptor territoriality.PLoSONE13(10):e0205799

Also, could you include a reference supporting those lines.

Thank you, we adjusted the language of this sentence to be more general and added the following citation

L 96. “For avian predators that may utilize a wide variety of prey items, structural resources may be a primary factor influencing habitat use [22].”

Greenwald DN, Crocker-Bedford DC, Broberg L, Suckling KF, Tibbitts T. A review of northern goshawk habitat selection in the home range and implications for forest management in the western United States. Wildlife Society Bulletin. 2005;33: 120–148. 

In the title and abstract you mentioned the term generalist predators but in the introduction only referred to avian predators, except in L. 96-97. This may be confused for readers. I suggest being consistently with one term for predators. In the objectives L. 148-155 you use avian predators not generalist predators. I suggest the use of generalist predators in the manuscript.

Thank you, we added language on L. 122 to indicate that when we refer to avian predators throughout the rest of the manuscript we are referring to our two generalist avian predator study species.

In recent years, increased abundance in sagebrush habitat of common ravens (Corvus corax) and red-tailed hawk (Buteo jamaicensis), two highly generalist avian predators (hereafter “avian predators”)

Methods

Please include references in the section of Raven and raptors counts.

Thank you, we added the following reference

L198. 46. Andersen, D.E. Survey techniques. In: Raptor research and management techniques. 2007. pp. 89-100.

L 198 How much time did you spend in stationary counts? This was equal in all transects? Please describe.

Thank you, we clarified these methods..

L. 195 10-minute observation periods

L199 – 200. Along each survey transect, the amount of time spent at stationary survey locations was consistent, but walking surveys between stationary survey locations varied based on terrain.

L 215 Please mention the ground squirrel species

Thank you, we moved our description of the ground squirrel species at our site from line 233 to line 216.

L225-226 The transects for songbirds was an extension of 1.200 m, you mentioned that transects for avian predators had 800 m with 3 points counts within the transect in L 194-195. This it is not clear.

Thank you, we clarified this sentence describing the placement of songbird survey stations along the avian predator survey transect:

L 227-229. Each survey transect for avian predators had three point counts stations placed at either end and in the middle of the survey transect.

L 233-235 “Belding’s ground squirrels (Urocitellus beldingi), which occur in large semi-colonial populations, are the most common ground squirrel at our study site.” This could be used in the first paragraph of prey abundance. L. 214-216.

See above

Discussion

L 555-560 I suggest being more cautious with these lines, the relationship between the study and the presence of the sage-grouse is not clear.

L. 562-564 Thank you, we added the following language about the limits for inference that our study provides.

“However, because we did not directly examine the impact of habitat-use by avian predators on sage-grouse demographics, future research on this topic will be an important next step for managers aiming to conserve populations of prey species.”

---

## [Editor Report · Decision Letter 2]

17 May 2023

Implications of tree expansion in shrubland ecosystems for two generalist avian predators .

PONE-D-22-29377R2

Dear Dr. Young,

We’re pleased to inform you that your manuscript has been judged scientifically suitable for publication and will be formally accepted for publication once it meets all outstanding technical requirements.

Kind regards,

Juan Manuel Pérez-García, PhD

Academic Editor

PLOS ONE

---

## [Editor Report · Acceptance letter]

23 May 2023

PONE-D-22-29377R2 

Implications of tree expansion in shrubland ecosystems for two generalist avian predators.

Dear Dr. Young:

I'm pleased to inform you that your manuscript has been deemed suitable for publication in PLOS ONE. Congratulations! Your manuscript is now with our production department. 

Kind regards, 

on behalf of

Dr. Juan Manuel Pérez-García 

Academic Editor

PLOS ONE